# ROYAL SOCIETY
# OPEN SCIENCE

# Research

nanotechnology/materials science

mangosteen extract, colloid, self-assembly

**Author for correspondence:**
Supason Wanichwecharungruang
e-mail: psupason@chula.ac.th

This article has been edited by the Royal Society of Chemistry, including the commissioning, peer review process and editorial aspects up to the point of acceptance.

# Water-dispersible unadulterated α-mangostin particles for biomedical applications

Jutamad Bumrung[1,2], Chanpen Chanchao[3],
Varol Intasanta[5], Tanapat Palaga[4]
and Supason Wanichwecharungruang[1,2]

[1]Department of Chemistry, Faculty of Science, [2]Center of Excellence in Advanced Materials and Biointerfaces, [3]Department of Biology, Faculty of Science, and [4]Department of Microbiology, Faculty of Science, Chulalongkorn University, Bangkok, Thailand
[5]National Nanotechnology Center, National Science and Technology Development Agency, Pathumthani, Thailand

ID SW, 0000-0002-2802-4341

α-Mangostin, the extract from pericarp of *Garcinia mangostana* L. or mangosteen fruit, has been applied in various biomedical products because of its minimal skin irritation, and prominent anti-inflammatory, antimicrobial and immune-modulating activities. Owing to its low water solubility, the particle formulations are necessary for the applications of α-mangostin in aqueous media. The particle formulations are usually prepared using surfactants and/or polymers, usually at a larger amount of these auxiliaries than the amount of α-mangostin itself. Here, we show the self-assembly of α-mangostin molecules into water-dispersible particles without a need of any polymers/surfactants. Investigations on chemical structure, crystallinity and thermal properties of the obtained α-mangostin particles, in comparison to the conventional α-mangostin crystalline solid, confirm no formation of the new compound during the particle formation and suggest changes in intermolecular interactions among α-mangostin molecules and significantly more hydroxyl functionality positioned at the particles' surface. The ability of the water suspension of the α-mangostin to inhibit the growth of *Propionibacterium acnes*, the acne-causing bacteria, is similar to that of the solution of the conventional α-mangostin in 5% dimethyl sulfoxide. Moreover, at 12.7 ppm in an aqueous environment of RAW 264.7 cell culture, α-mangostin suspension exhibits five times higher anti-inflammatory activity than the conventional α-mangostin solution, with the same acceptable cytotoxicity of less than 20% cell death.

# 1. Introduction

α-Mangostin, a yellow bitter prenylated xanthone, is the major component in the water-insoluble yellowish material found in the pericarp of *Garcinia mangostana* L. or mangosteen fruit. The pericarp of this fruit has been used as a medicinal agent by southeast Asians for centuries. During the last few decades, scientific researchers have reported that α-mangostin possesses various biological activities including anti-inflammatory [1], antibacterial [2], antifungal [3], antiviral [4], antimycobacterial, antiallergic [5] and antitumoral activities [6]. Researches have also demonstrated impacts of α-mangostin on activities of various enzymes, e.g. α-glucosidase [7], aromatase [8], HIV-1 protease [9], inhibitor kB kinase [10], quinone reductase [11], sphingomyelinase [12], topoisomerase [13] and several other protein kinases [14], as well as the binding to histamine $H_1$ [15] and 5-hydroxytryptamine$_{2A}$ receptors [16]. Also, studies using α-mangostin have shown a positive impact on hyperglycemia, atherosclerotic lesions and skin inflammation in whole animals and human volunteers [7,17–19].

Low solubility in water of α-mangostin ($0.2 \pm 0.2$ μg ml$^{-1}$ at room temperature) [20] has prevented its application in beverages, whereas conventional applications in topical formulations require the presence of either organic solvents such as ethanol, or surfactants [18,21]. Skin irritation against organic solvents and small surfactant molecules, and instability of the small surfactant-based particles are well-known drawbacks [22].

Another reported technological approach to use α-mangostin in water media is through the entrapment of the material into water-dispersible polymer-based particles. We previously demonstrated the use of ethyl cellulose to make water-dispersible α-mangostin-loaded nanoparticles [23] and showed hair follicle entrapment and sebaceous gland deposition of the particles which led to effective acne vulgaris treatment [19,24,25]. Several other water-dispersible particles that contain α-mangostin have also been reported [26–28]. Although the polymer-based particles are more stable than the surfactant-based micellar and vesicular particles and can withstand massage action required for hair follicle entrapment [29], the polymers themselves take up mass and are a burden for the body to eliminate. Reported polymeric particles with more than 35% loading of α-mangostin are rare [19,23–28]. Because α-mangostin is quite a stable compound [30], the purpose of encapsulation is usually better for application in aqueous media, not for the improvement of its chemical stability. As a result, a strategy that can reduce or even abandon the use of unnecessary adulterated materials should be better for health application of α-mangostin.

Assemblies into water-dispersible nano-aggregates of conjugates of hydrophobic drug molecules containing hydrophilic oligomers, or complexes between hydrophilic peptide and small hydrophobic drug molecules have been reported in the past decades [31–34]. Those assemblies were predictable as the drug conjugate structures resemble the surfactant molecules with the hydrophobic head (drug molecule) and hydrophilic tail parts (auxiliaries). The long hydrophilic oligomeric parts in the drug conjugates, again, take up mass and cause elimination burden to the body. A big step came around 2015 when researchers reported the construction of water-dispersible particles through the assembling of small hydrophobic drug molecules with no long hydrophilic tail in their structure. For example, the self-assembly of small hydrophobic camptothecin dimers with no hydrophilic tail section in the structure, into water-dispersible nanospheres, has been demonstrated [35]. Many other small hydrophobic drugs with small chemical alteration or conjugation have recently been designed and self-assembled into water-dispersible particles [36–38]. These reports have established the research in the supramolecular assembly of pure drugs into self-delivery systems [39,40]. We have noticed that the strategic mechanism commonly used to induce these assemblies is the staggering of aromatic rings, i.e. drug molecules reported capable of assembling into such particles usually possess an aromatic ring system [31–43].

To the best of our knowledge, the reported pure drug assembly strategy still requires some chemical alteration or derivatization of the drug molecules. Here, we show the assembly of pure unmodified α-mangostin into particles that can stably disperse in water with no need of any excipient. This paper reports a systematic study on how factors such as drug concentration and type and concentration of solvent influence the particle formation. We use various spectroscopic techniques to probe into the molecular arrangement of the water-dispersible α-mangostin particles obtained under different conditions. Finally, we also report here the biological activity, the *in vitro* anti-*Propionibacterium acnes* (*P. acne*) and anti-inflammatory activities, of the aqueous suspension of the prepared α-mangostin particle in comparison to those of the conventional solution of α-mangostin in 5% dimethyl sulfoxide (DMSO).

# 2. Material and methods

## 2.1. Prepareration of mangostin particles

α-Mangostin (94% α-mangostin, Welltech Biotechnology Co., Ltd., Thailand) was dissolved in ethanol. Then water was slowly dropped at the rate of 2.5 ml min⁻¹ into the α-mangostin solution with continuous stirring at 1000 rpm, and particles were formed. After that, ethanol was removed under vacuum at 30°C to give a yellow suspension. The suspension was freeze-dried (Labconco Corporation, Kansas, MI, USA) to obtain α-mangostin particle (MG-E) powder. The effect of α-mangostin concentration and ethanol/water ratio on particle formation was investigated (see the electronic supplementary material, table S1).

The particle assembly was also carried out using propanol, butanol, methanol, acetone and DMSO in place of ethanol, and the respective products were freeze-dried. $^1$H-nuclear magnetic resonance (NMR) spectrophotometric analyses were performed on samples (20 mg ml⁻¹ in dimethyl sulfoxide-d6 (DMSO-d6)) using JNM-ECZR 500 MHz, JEOL, Japan. Dry particles were subjected to scanning electron microscopic (SEM; JSM-IT100 JEOL, Tokyo, Japan), attenuated total reflection Fourier transform-infrared (ATR FT-IR) spectrophotometric (Nicolet 6700, Thermo Electron Corporation, Madison, WI, USA), powder X-ray diffraction (XRD; scanning rate 15°–60° in the $2\theta$ range, scan speed 5° min⁻¹ in 0.02° step size, voltage 40 kV, electronic current 30 mA, Rigaku X-ray diffractometer DMAX 2200/Ultima, Japan), thermogravimetric (scanned from 50°C to 80°C at the heating rate of 20°C min⁻¹ under the nitrogen, Netzsch STA 449 F1, Germany) and X-ray photoelectron spectroscopic (XPS; Al K$\alpha$ radiation, CasaXPS software, Kratos X-ray photoelectron spectrometer, Axis-Ultra DLD XSAM800) analyses. Particle sizes and zeta potentials of the aqueous suspension of the particles were acquired by dynamic light scattering (DLS) technique using a zetasizer nano-series instrument (Zs; Malvern Instruments, United Kingdom). The sizes of dry particles were obtained from SEM images using *ImageJ* software.

## 2.2. Biological activity testing

### 2.2.1. Anti-acne activity

The ability of the MG-E to inhibit *P. acne* was determined by measuring a clear inhibition zone using an agar disc diffusion method [44]. *Propionibacterium acne* (ATCC 6919, Department of Medical Sciences, Ministry of Public Health, Thailand) was incubated in brain heart infusion broth (3.8% w/v, HiMedia Laboratory, India) at 37°C for 72 h under anaerobic conditions. *Propionibacterium acne* suspension was adjusted to 0.5 McFarland standard to obtain approximately cell density $1 \times 10^8$ CFU ml⁻¹. Brain heart infusion agar (5.5% w/v, 20 ml, HiMedia Laboratory, India) was used as an agar base. Then prepared *P. acne* was coated on the agar base. A sterile paper disc (6 mm) was placed on the agar. Each sample (20 µl) was dropped on the disc. The samples included the MG-E suspended in water and the solution of the original α-mangostin in 5% DMSO in water, at the concentrations of 1.25, 2.5, 5.0, 10.0 and 20.0 mg ml⁻¹. After that, plates were incubated at 37°C for 72 h under anaerobic conditions, and the clear zone was measured in millimetres.

### 2.2.2. Anti-inflammatory activity and cell viability

*In vitro* anti-inflammatory property of tested samples was investigated with RAW 264.7 cells (ATCC, Manassas, Virginia, USA) using nitric oxide (NO) assay [45]. The samples include the MG-E suspended in water and the solution of the original α-mangostin in 5% DMSO in water, at the concentrations of 7.12, 9.50, 12.66, 16.88, 22.50 and 30.00 ppm. To ensure that the obtained anti-inflammatory activity was unrelated to the cytotoxicity of the tested samples, effects of samples on cell viability were evaluated by 3-(4,5-dimethylthiazol-2-yl)-2,5-diphenyltetrazolium bromide (MTT) assay [46].

#### 2.2.2.1. Cell culture conditions and cell treatment

RAW 264.7 cells were cultured in complete media (87% v/v Dulbecco's modified Eagle's medium (Hyclone, Utah, USA), 1% v/v sodium pyruvate, 1% v/v 4-[2-hydroxyethyl]-1-piperazineethanesulfonic acid (Hyclone, Utah, USA), 1% v/v gentamicin and 10% v/v fetal bovine serum (Hyclone, Utah, USA)), under a humidified atmosphere of 5% $CO_2$ at 37°C for 72 h. Then the cells were seeded in 96-well plate ($1 \times 10^4$ cell well⁻¹) overnight. After that, the supernatant was removed and replaced with sample

(50 µl well$^{-1}$) and incubated for 1 h. Next, complete media containing lipopolysaccharide (200 ng ml$^{-1}$, Sigma-Aldrich, Streinheim, Germany) and gamma interferon (20 ng ml$^{-1}$, Sigma-Aldrich, Streinheim, Germany) was added (50 µl well$^{-1}$), and the cells were incubated for 24 h. The treated cells were divided into two parts for NO assay (2.2.2.2) and MTT assay (2.2.2.3).

### 2.2.2.2. Nitric oxide assay

Amount of nitrite is an indicator of NO synthesis. Here, the amount of nitrite was measured in the culture medium by a colorimetric method using Griess reagent [45]. In the method, the supernatant (50 µl) of the treated cells from §2.2.2.1 were mixed with Griess reagent (0.1% N-1-napthylethylenediamine dihydrochloride (NED, Sigma-Aldrich, Streinheim, Germany) and 1% sulfanilamide (Merck, Darmstadt, Germany) in 5% phosphoric acid). The mixture was then subjected to UV absorption spectroscopic analysis at 540 nm. The nitrile concentration was determined with the aid of the standard curve constructed from standard solutions of sodium nitrite in the culture medium. Water was used as a control. Per cent NO production was calculated as follows:

$$\% \text{ NO production} = \frac{\text{nitrite concentration of the tested sample}}{\text{nitrite concentration of water}} \times 100.$$

### 2.2.2.3. MTT assay

The evaluation of cell viability was based on the mitochondrial dehydrogenase activity in the living cells to reduce MTT into a purple formazan product which can be measured by UV absorption spectroscopy [46]. Here, the remaining supernatant of the cells on the plate from §2.2.2.1 was removed and replaced with complete media (100 µl well$^{-1}$) and MTT solution (5 mg ml$^{-1}$ in phosphate buffered saline pH = 7.4, 10 µl well$^{-1}$). Then cells were incubated in a humidified atmosphere at 37°C for 4 h. After that, DMSO was added (200 µl well$^{-1}$) to dissolve purple formazan. After completely dissolving, the absorbance was measured at 540 nm. Water was used as a control and complete media was used as a blank. Cell viability was calculated as follows:

$$\% \text{ cell viability} = \frac{\text{abs of sample} - \text{abs of media}}{\text{abs of water} - \text{abs of media}} \times 100.$$

# 3. Results and discussion

By slowly introducing water into the ethaolic solution of α-mangostin, particulate formation was observed as the clear solution turned cloudy. The dry particles, obtained by removing ethanol from the suspension by vacuum and then subjecting the aqueous suspension to freeze drying, could disperse well in water. We systematically studied the effect of α-mangostin concentrations (0.5, 1.0, 5.0, 10.0 and 30.0% w/v, nos 1–5 of the electronic supplementary material, table S1) on the particle formation. The result clearly showed that at the concentration range of 0.5–5.0% w/v of α-mangostin in the starting ethanolic solution, only water-dispersible particles were obtained, no settling down precipitate was observed. The diameter of the formed particles increased with increasing concentration of the starting α-mangostin in ethanol. The biggest particles that still showed good water dispensability possess diameters of less than 10 µm (figure 1$a$i–$a$iii). When the starting α-mangostin concentration reached 10% w/v, observable precipitates (not dispersible in water) started to appear. Higher concentration (30% w/v) led to more precipitates (figure 1$b$i). We explain these observations as follows. When water was slowly added to the ethanolic solution of α-mangostin, phase separation of α-mangostin occurred, as α-mangostin molecules are poorly soluble in water. Because the addition of water was slow and the concentration of α-mangostin was not too high, these molecules arranged themselves in such a way that the hydrophobic part was away from water and the hydrophilic part interacted with the added water. At higher starting α-mangostin concentrations, more α-mangostin molecules were surrounding each growing particle during the water addition, causing faster deposition of the molecules onto each growing particle, resulting in more α-mangostin molecules being integrated into each particle, thus a bigger sized particle was obtained. This faster deposition also resulted in less time for molecular orientation to take place for the depositing molecules. With less hydrophilic moieties properly facing out at the particles' surface, the particle's surface became more hydrophobic and resulted in the particle being even more attractive to α-mangostin molecules. These contribute to the random aggregation of α-mangostin molecules into large precipitates.

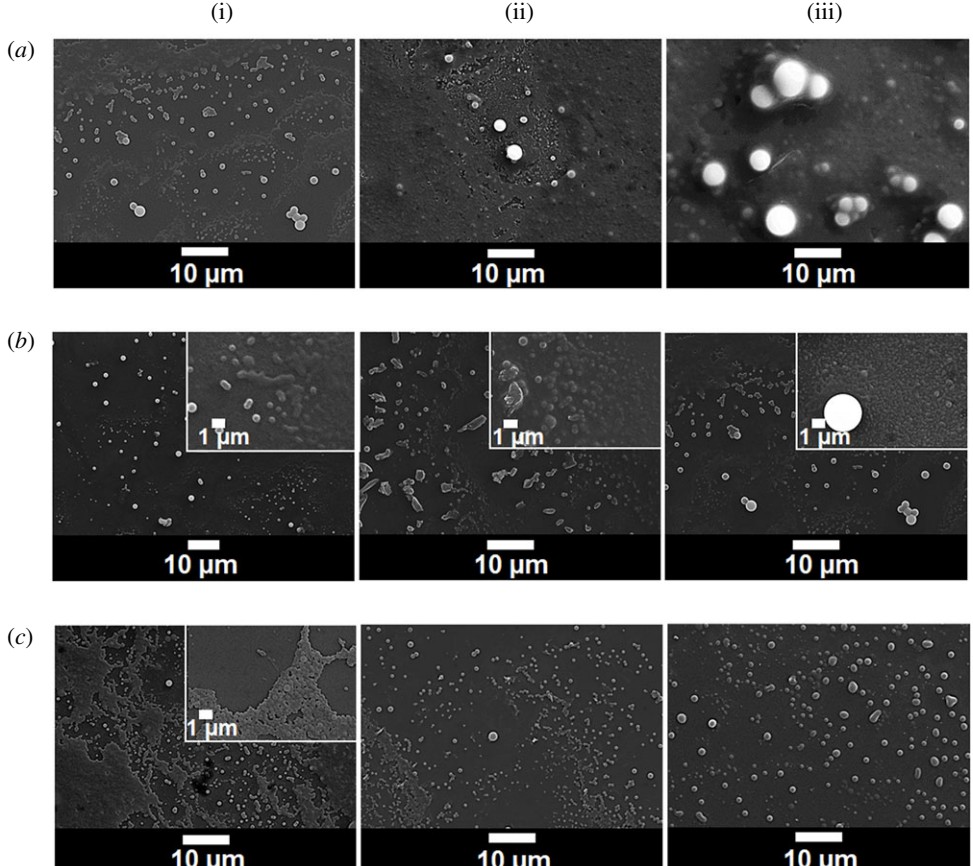

**Figure 1.** SEM images of α-mangostin particles prepared under various conditions: (*a*) α-mangostin concentrations of 0.5% (i), 1% (ii), and 5% (iii); (*b*) ethanol to water ratios of 0.5% (i), 5% (ii), and 20% (iii); and (*c*) 50% solvent to water ratio when the solvent is ethanol (i), propanol (ii), and butanol (iii).

We prepared α-mangostin particles at various final ethanol to water ratios (nos 6–9 of the electronic supplementary material, table S1), dried the obtained suspension, then analysed the morphology of the products. From the SEM images (figure 1 and electronic supplementary material, figure S1), the diameter of the particles decreased as the ethanol to water ratio increased. The particles made at 50% ethanol in water (MG-E) gave the most uniform particle with an average dry size of 312.17 ± 55.11 nm. Under this condition, no precipitate was observed. It should be noted here again that the mixture was totally dried at the end of the process; therefore, no α-mangostin could be left out in the liquid part of the mixture as soluble molecules. All α-mangostin molecules in the system had assembled into water-dispersible particles.

We investigated the effect of replacing ethanol with other water-miscible solvents. Among propanol, butanol, methanol, acetone and DMSO, only propanol and butanol gave water-dispersible particles of α-mangostin. SEM images indicate that these dry particles are spherical (psupason@chula.ac.th) with the diameters of 324.35 ± 71.45 and 467.97 ± 188.51 nm, for those made with propanol (MG-P) and butanol (MG-B), respectively. Other solvents gave precipitates.

Particle size measurements of MG-E in water by DLS showed unimodal distribution with an average diameter of 370.19 ± 50.00 nm (electronic supplementary material, figure S2), which was significantly bigger than the size of the dry particles observed by SEM (312.17 ± 55.11 nm). The MG-P and MG-B suspensions in water, however, showed bimodal distributions (electronic supplementary material, figure S2). We speculated that the bimodal distribution was possibly a result of the breakup of large particles during the organic solvent evaporation at the end of the process. It was possible that the evaporation of propanol or butanol that had been trapped in the particles might cause particle destabilization and break-up. This did not happen when ethanol was used because ethanol is more hydrophilic and thus entrapment was probably not as significant. In addition, the boiling point of ethanol is the lowest among the three solvents; therefore, evaporation could take place easily. By contrast, the trapped propanol and butanol which possess higher boiling points probably built up

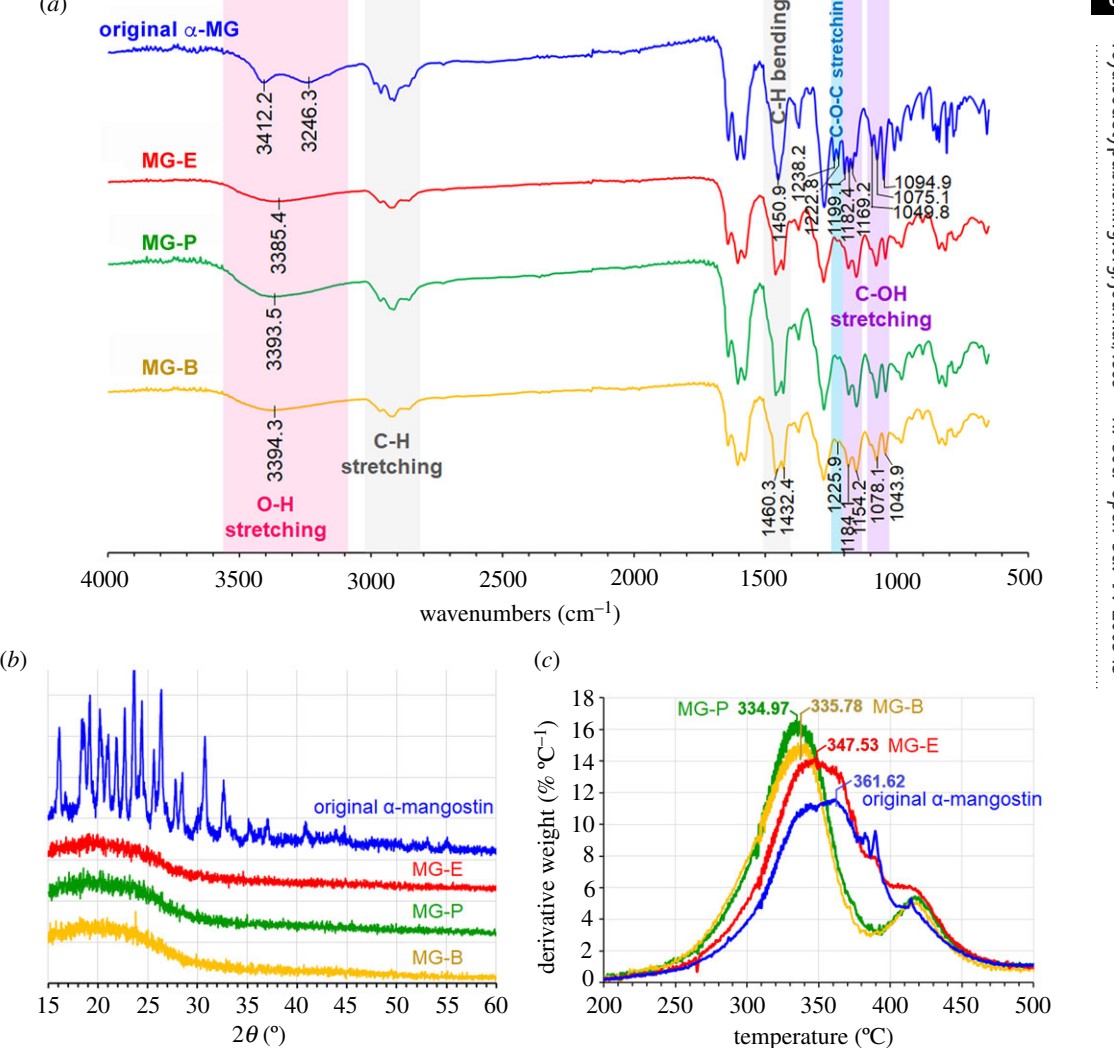

**Figure 2.** Structural analyses of original α-mangostin crystalline solid, MG-E (mangostin particles made in 50% aqueous ethanol), MG-P (mangostin particles made in 50% aqueous propanol), and MG-B (mangostin particles made in 50% aqueous butanol): (a) IR spectra, (b) XRD patterns, and (c) thermograms.

vapour pressure inside the particles at higher temperatures where intermolecular bonds among α-mangostin were weakening, resulting in particle disintegration. Further study is needed to make a better understanding of the mechanism of the particle formation and their size distribution.

The average zeta potential values of MG-E, MG-P and MG-B, in water, are $-34.1 \pm 0.72$, $-30.1 \pm 0.21$ and $-32.9 \pm 1.56$ mV, respectively. These zeta potential values agree well with the observation that water dispersions of the three α-mangostin particles are stable.

To make sure that no chemical transformation took place on the α-mangostin structure during the preparation of the particles, we dissolved the obtained MG-E in DMSO-d6 and acquired their [1]H NMR spectrum. Indeed, no chemical transformation took place during the particle formation, the [1]H NMR spectrum of the MG-E particles is identical to that of the original bulk α-mangostin (electronic supplementary material, figure S3). This also applies to the [1]H-NMR spectra of MG-P and MG-B.

Interestingly, significant differences in the FT-IR spectrum of the original α-mangostin and those of the α-mangostin particles (MG-E, the MG-P and the MG-B) could be clearly observed (figure 2a). It should be noted here that, unlike [1]H NMR that all intermolecular forces within the particles did not exist owing to the complete dissolution of the material before analysis, FT-IR analyses were performed directly with the material in their self-assembled particulate state. The O-H stretching vibration of the original α-mangostin shows up as the double broad peaks at 3412.2 and 3246.3 cm$^{-1}$ whereas that of the three α-mangostin particles appears as a single broad peak at 3385.4 cm$^{-1}$. The C-O stretching vibration of vinyl ether of the original α-mangostin appears as the double peaks at 1238.2 and 1222.8 cm$^{-}$

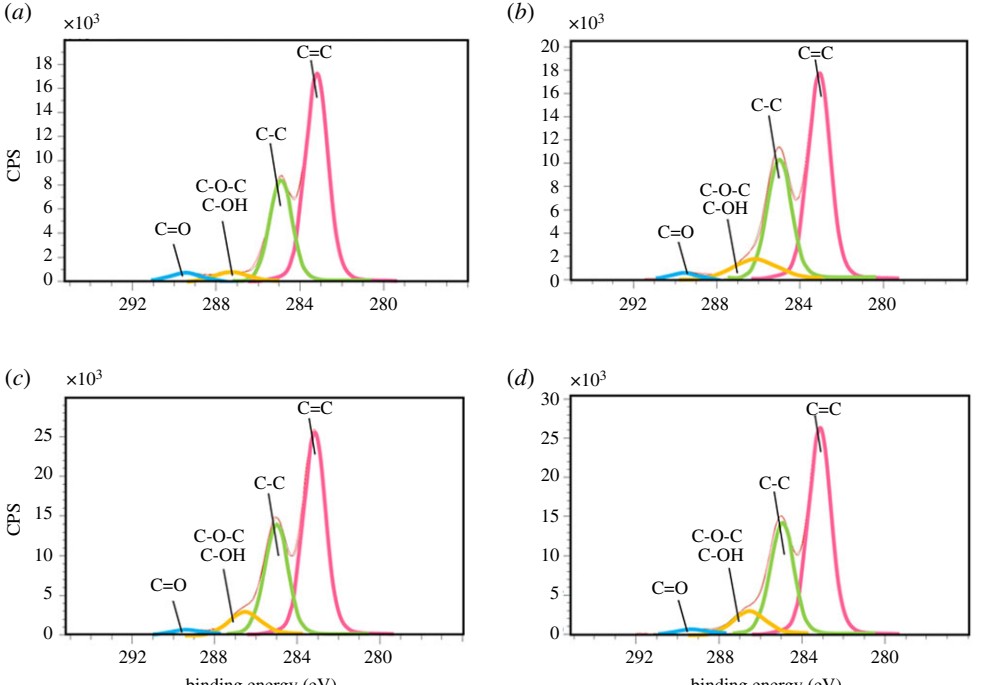

**Figure 3.** Deconvoluted C-1 s spectra of (*a*) original α-mangostin, (*b*) MG-E, (*c*) MG-P, and (*d*) MG-B.

[1] whereas that of the three α-mangostin particles appears as the single peak at 1226.8 cm$^{-1}$. This indicates that the self-assembled α-mangostin molecules in the particles probably exist in more than one configuration. The stretching vibration of the C-O bond of the -C-O-H functionality appears as two sets of the triple peaks at 1199.1, 1182.4, 1169.2 cm$^{-1}$ and 1094.9, 1075.1, 1049.9 cm$^{-1}$ for the original α-mangostin, whereas that of the three mangostin particles appears as two sets of the double peaks at 1184.1, 1154.2 cm$^{-1}$ and 1078.7, 1044.1 cm$^{-1}$. In addition, vibrational absorptions from functionalities associated with the aromatic π-π bonding of the original raw material also differ from those of the three particles, i.e. C=C stretching of the aromatic ring in the original raw material absorbs at 1607.3 and 1581.4 cm$^{-1}$, whereas that of the three particles absorbs at 1605.1 and 1580.4 cm$^{-1}$, and the C=O stretching of the aromatic ketone in the original raw material absorbs at 1640.1 cm$^{-1}$, whereas that of the three particles absorbs at 1643.4 cm$^{-1}$. Because all the tested materials are pure α-mangostin with the same molecular structure, the differences in the FT-IR spectra imply different configuration settings of the molecules and different intermolecular interactions. The information indicates that H-bonding and π-π stacking among α-mangostin molecules in the original α-mangostin differ from those in the three α-mangostin particles.

The XRD pattern of the original α-mangostin reveals crystalline character with the 2θ at 16.1, 18.5, 19.2, 20.2, 21.0, 21.8, 22.7, 23.6, 24.4, 25.6, 26.3, 27.7, 28.5, 30.8 and 32.6° (figure 2*b* blue line). By contrast, the XRD patterns of MG-E, MG-P and MG-B (figure 2*b* red, green and yellow) contain only a single broad peak, clearly indicating amorphous structure. These results confirm that the assembly of the α-mangostin molecules into water-dispersible α-mangostin particles results in different molecular packing from that of the original water-immiscible α-mangostin solid.

Thermograms of the original α-mangostin, MG-E, MG-P and MG-B are not identical (figure 2*c*). The original α-mangostin, MG-E, MG-P and MG-B show the maximum decomposition temperature at 361.62, 347.53, 334.97 and 335.78°C, respectively. These results indicate different molecular packing of α-mangostin molecules in the four materials. The molecular packing of the α-mangostin particles is less tight compared to that of the original α-mangostin. In addition, intermolecular forces of the MG-P and MG-B (particles prepared using propanol and butanol as solvent, respectively) are a little weaker than that of the MG-E (particles prepared using ethanol as solvent). This lower thermal stability of the water-dispersible particles as compared to that of the original α-mangostin corresponds well to the disappearance of the crystallinity of particles observed in the above XRD patterns and the difference in the intermolecular interactions implied by their FT-IR spectra. All these observations also agree with our proposed model of solvent entrapment in the particles. The presence of solvent molecules at the inside of the particles during the assembling process would prevent crystalline packing inside the particles and decrease thermal stability.

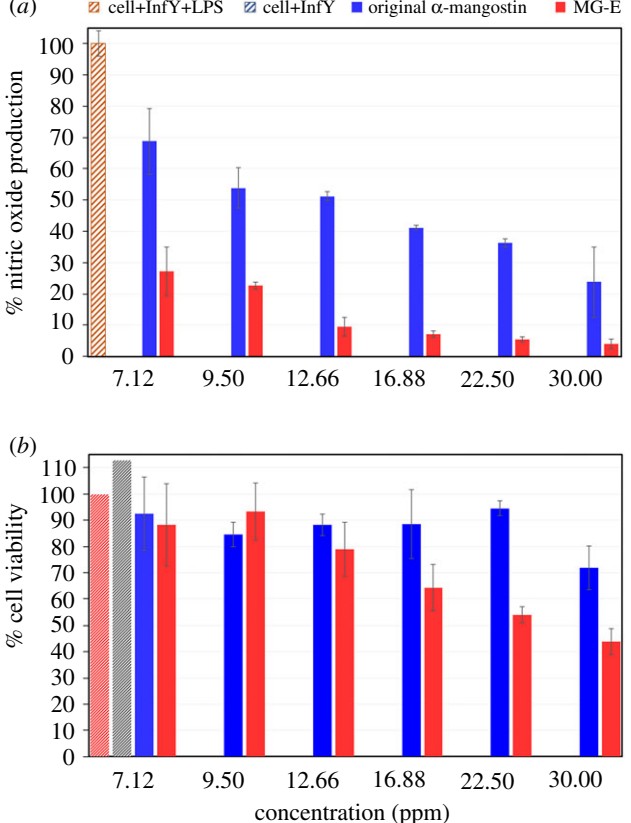

**Figure 4.** Anti-inflammatory activity of the original α-mangostin solid and MG-E: (*a*) % NO production and (*b*) % cell viability.

Next we probed the surface of the particles with the XPS analysis. Although the C1 s spectra of the original α-mangostin and the MG-E (electronic supplementary material, table S2 and figure 3) show the same deconvoluted peaks at the binding energy of 283.2, 285.0, 286.9 and 289.4 eV which correspond to C=C, C-C, C-O-C/C-OH, and C=O bonds, respectively, the proportion of the peak areas are different among the spectra of the four materials. The peak area of C-O-C and C-OH (at 286.9 eV) increase from 4.70% in the original α-mangostin to 10.50, 8.66 and 9.14% in the MG-E, MG-P and MG-B, respectively. Because the average depth of analysis for an XPS measurement is approximately 5 nm, such increases in the peak area of C-O-C and C-OH moieties indicate more accumulation of hydroxyl moieties at the surface of the MG-E particles followed by MG-P, MG-B. The original α-mangostin contains the least C-O-C and C-OH moieties on its surface. The implication from the XPS results which is that the α-mangostin particles contain more hydrophilic moieties at the surface as compared to the original α-mangostin solid, agrees well with their water dispersibility. It is likely that these hydrophilic functionalities at the surface of the particles form hydrogen bonding to water molecules and thus make the particles dispersible in water medium. We speculate that the enthalpy compensation from the hydrogen bonding interactions with water molecules at the particle's surface is the key driving force that propels the system into the formation of α-mangostin particles even though the colloidal particles contain weaker intermolecular forces among α-mangostin molecules inside the particles as compared to that of the original α-mangostin solid.

Next, we tested the ability of the original α-mangostin and the MG-E to inhibit the growth of *P. acne*, the bacteria that cause acne. The test was carried out by dissolving the original α-mangostin solid in DMSO and then diluting it with the working medium, making sure that there was no precipitate formation. By contrast, the MG-E was tested with no organic solvent, only water was used to disperse the material. The clear inhibition zones against *P. acne* of the original α-mangostin and MG-E are comparable (electronic supplementary material, table S3 and figure S4). This result indicates that the α-mangostin colloidal particles in water possess the same anti-*P. acne*, as that of the original α-mangostin in organic solvent.

Using the RAW 264.7 cell line, we tested the anti-inflammatory activities, another well-known biological activity of α-mangostin, of the MG-E via the NO assay. The original α-mangostin was first dissolved in

DMSO and then diluted with water (to 5% DMSO), whereas the MG-E was directly dispersed in water. Here, anti-inflammatory activity of the tested sample was determined as the ability of the sample to suppress the NO production, which was calculated from the amount of the nitrite accumulated in the culture supernatants. The % NO production (figure 4a) indicates that the anti-inflammatory activity of MG-E suspension is significantly higher than that of the original α-mangostin solution. The MTT assay indicates that the MG-E at concentrations up to 12.7 ppm is not toxic to the tested cells (greater than 80% cell viability, figure 4b). To our surprise, at this highest non-toxic concentration of 12.7 ppm, the MG-E suspension was five times more potent in inhibiting the inflammatory reaction in the cells than the original α-mangostin solution. Thus, it can be concluded that the α-mangostin particles obtained from this self-assembly process possess better anti-inflammatory activity than the original α-mangostin, and this obtained anti-inflammatory activity was not owing to the cytotoxicity effect of α-mangostin. We speculate that the colloidal state of the material probably helps improve the efficiency of the uptake of the sample by RAW 264.7 cells. In other words, the MG-E colloids probably can get into the RAW 264.7 cells more effectively than α-mangostin in the true solution state.

## 4. Conclusion

Here, we have shown that through the use of water and appropriate water-miscible solvents, self-assembly of the water-insoluble α-mangostin into the water-dispersible particles of pure α-mangostin could be induced. The prepared pure α-mangostin particles can be easily dispersed into a stable suspension in water with no need of any auxiliary agent. FT-IR, XRD spectra and thermograms indicate that molecular packing and intermolecular interactions of the α-mangostin molecules in the particles are different from those in the original solid. The original solid seems to possess stronger intermolecular attractions as witnessed through its better thermal stability. The alteration of the intermolecular forces in the prepared particles revolves around the aromatic hydroxyl groups and the aromatic π-π stacking as suggested from the FT-IR spectra. The difference in molecular packing of the prepared particles from that of the original solid is also reflected in the loss of crystallinity upon particle formation as shown in their XRD spectra. Water dispersibility of the particles could be explained at the molecular level through the observation, via XPS, of the significantly higher abundance of polar hydroxyl groups at the particle surface, as compared to that at the surface of the original solid. We speculate that the enthalpy compensation from the hydrogen bonding interactions between water molecules and these hydroxyl groups at the particle's surface is the driving force for the system to favour the particle formation regardless of the weaker intermolecular forces among α-mangostin molecules inside the particles. When tested by the disc diffusion assay, the aqueous suspension of the α-mangostin particles could inhibit the growth of P. acne at a similar level to that of the original α-mangostin solution (used with some DMSO to aid solubility). When tested by the NO assay in the RAW 265.7 cell line at a concentration that was not toxic to the cells, the aqueous suspension of the α-mangostin particles showed five times higher anti-inflammatory activity than the original α-mangostin in the solution state. We speculate that the α-mangostin particles probably could be more effectively taken up into the cells as compared to α-mangostin in the solution state.

Data accessibility. This article has no additional data.

Authors' contributions. J.B. performed all the experiments; C.C. helped design the P. acne experiment; V.I. helped with the experimental design; T.P. helped designing cellular experiments; S.W. designed the whole work and wrote the manuscript.

Competing interests. We declare we have no competing interests.

Acknowledgements. This work was funded by the National Nanotechnology Center (NANOTEC), NSTDA, Ministry of Science and Technology, Thailand, through its programme of Research Network NANO- TEC (RNN), and the Center of Excellence in Advanced Materials and Biointerfaces, Chulalongkorn University.

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
