## [Reviewer comments · Royal Society Open Science]

Review History

RSOS-200543.R0 (Original submission)

Review form: Reviewer 1

Is the manuscript scientifically sound in its present form?

Yes

Are the interpretations and conclusions justified by the results?

Yes

Is the language acceptable?

Yes

Do you have any ethical concerns with this paper?

No

Have you any concerns about statistical analyses in this paper?

No

Recommendation?

Accept with minor revision (please list in comments)

Comments to the Author(s)

Please see attached (Appendix A).

Decision letter (RSOS-200543.R0)

Dear Dr Wanichwecharungruang:

Title: Water-dispersible unadulterated α -mangostin nanoparticles for biomedical applications
Manuscript ID: RSOS-200543

Thank you for submitting the above manuscript to Royal Society Open Science. On behalf of the Editors and the Royal Society of Chemistry, I am pleased to inform you that your manuscript will be accepted for publication in Royal Society Open Science subject to minor revision in accordance with the referee suggestions. Please find the reviewers' comments at the end of this email. The Associate Editor was not able to find a second reviewer but they have provided some comments of their own (please see below) to avoid any further delay.

The reviewers and handling editors have recommended publication, but also suggest some minor revisions to your manuscript. Therefore, I invite you to respond to the comments and revise your manuscript.

Because the schedule for publication is very tight, it is a condition of publication that you submit the revised version of your manuscript before 19-Aug-2020. Please note that the revision deadline will expire at 00.00am on this date. If you do not think you will be able to meet this date please let me know immediately.

- 1) A text file of the manuscript (tex, txt, rtf, docx or doc), references, tables (including captions) and figure captions. Do not upload a PDF as your "Main Document".
- 2) A separate electronic file of each figure (EPS or print-quality PDF preferred (either format should be produced directly from original creation package), or original software format)
- 3) Included a 100 word media summary of your paper when requested at submission. Please ensure you have entered correct contact details (email, institution and telephone) in your user account

4) Included the raw data to support the claims made in your paper. You can either include your data as electronic supplementary material or upload to a repository and include the relevant doi within your manuscript

5) All supplementary materials accompanying an accepted article will be treated as in their final form. Note that the Royal Society will neither edit nor typeset supplementary material and it will be hosted as provided. Please ensure that the supplementary material includes the paper details where possible (authors, article title, journal name).

Kind regards,
Dr Laura Smith
Publishing Editor, Journals

On behalf of the Subject Editor Professor Anthony Stace and the Associate Editor Dr Nadia Martinez Villegas.

RSC Associate Editor:
Comments to the Author:
Strictly speaking, the size of the alfa-mangostin particles obtained in this study falls in the range of fine particles. Therefore, the use of the nano suffix should be removed from this manuscript. Instead, the use of fine particles or particles is suggested.

RSC Subject Editor:
Comments to the Author:
(There are no comments.)

Reviewer comments to Author:
Reviewer: 1

Comments to the Author(s)
Please see attached.

Author's Response to Decision Letter for (RSOS-200543.R0)

See Appendix B.

RSOS-200543.R1 (Revision)

Review form: Reviewer 1

Is the manuscript scientifically sound in its present form?

Yes

Are the interpretations and conclusions justified by the results?

Yes

Is the language acceptable?

Yes

Do you have any ethical concerns with this paper?

No

Have you any concerns about statistical analyses in this paper?

No

Recommendation?

Accept with minor revision (please list in comments)

Comments to the Author(s)

Looks great! Only 2 minor comments:

1. Fig. 2 caption (page 4): change it to "Structural analyses of original α -mangostin crystalline solid, MG-E (mangostin particles made in 50% aqueous ethanol), MG-P (mangostin particles made in 50% aqueous propanol) and MG-B (mangostin particles made in 50% aqueous butanol): (a) IR spectra, (b) X-ray diffraction patterns and (c) thermograms."
2. Fig. 4 caption (page 6): This must be Fig. 4 (not Fig. 2), correct accordingly.

Decision letter (RSOS-200543.R1)

Dear Dr Wanichwecharungruang:

Title: Water-dispersible unadulterated α -mangostin particles for biomedical applications
Manuscript ID: RSOS-200543.R1

Thank you for submitting the above manuscript to Royal Society Open Science. On behalf of the Editors and the Royal Society of Chemistry, I am pleased to inform you that your manuscript will

be accepted for publication in Royal Society Open Science subject to minor revision in accordance with the referee suggestions. Please find the reviewers' comments at the end of this email.

The reviewers and handling editors have recommended publication, but also suggest some minor revisions to your manuscript. Therefore, I invite you to respond to the comments and revise your manuscript.

Because the schedule for publication is very tight, it is a condition of publication that you submit the revised version of your manuscript before 24-Sep-2020. Please note that the revision deadline will expire at 00.00am on this date. If you do not think you will be able to meet this date please let me know immediately.

Kind regards,
Dr Laura Smith
Publishing Editor, Journals

On behalf of the Subject Editor Professor Anthony Stace and the Associate Editor Dr Nadia Martinez Villegas.

RSC Associate Editor:
Comments to the Author:
Please attend the comments provided by the reviewer.

RSC Subject Editor:
Comments to the Author:
(There are no comments.)

Reviewer comments to Author:
Reviewer: 1

Comments to the Author(s)
Looks great! Only 2 minor comments:

1. Fig. 2 caption (page 4): change it to "Structural analyses of original α -mangostin crystalline solid, MG-E (mangostin particles made in 50% aqueous ethanol), MG-P (mangostin particles made in 50% aqueous propanol) and MG-B (mangostin particles made in 50% aqueous butanol): (a) IR spectra, (b) X-ray diffraction patterns and (c) thermograms."
2. Fig. 4 caption (page 6): This must be Fig. 4 (not Fig. 2), correct accordingly.

Author's Response to Decision Letter for (RSOS-200543.R1)

See Appendix C.

Decision letter (RSOS-200543.R2)

Dear Dr Wanichwecharungruang:

Title: Water-dispersible unadulterated α -mangostin particles for biomedical applications
Manuscript ID: RSOS-200543.R2

It is a pleasure to accept your manuscript in its current form for publication in Royal Society Open Science. The chemistry content of Royal Society Open Science is published in collaboration with the Royal Society of Chemistry.

On behalf of the Subject Editor Professor Anthony Stace and the Associate Editor Dr Nadia Martinez Villegas.

RSC Associate Editor
Comments to the Author:
(There are no comments.)

Reviewer(s)' Comments to Author:

Appendix A

In this work, the authors prepared water-dispersible α -mangostin particles using a straightforward approach of the slow addition of water into an ethanolic solution of α -mangostin without the use of surfactants or polymers. They also systematically studied the effect of α -mangostin concentration, ethanol/water ratio, as well as the type of organic solvent on the size of α -mangostin particles. They finally characterized particles using a variety of techniques (e.g., FTIR, XRD, TGA, XPS, SEM, DLS) and tested their anti-acne and anti-inflammatory activities.

This work is suitable for publication in RSOS, and it can be considered for publication after addressing the following comments and questions:

1. In **Title** (page 3/14), the authors used the word “*nanoparticles*”. Nanoparticles are routinely defined as particles with sizes in nanoscale (i.e., **between 1 and 100 nm**), exhibiting properties that are not found in their bulk counterparts. According to the paper, the average sizes of particles obtained from SEM images are **above 300 nm**, so **I would suggest changing the word “nanoparticles” in the title and the whole manuscript to either “colloidal particles” or “particles”**. I also suggest mentioning the average sizes of particles in the Abstract (page 3/14).
2. In **Abstract** (line 8, page 3/14), the authors mention “*no chemical transformation*”; what does this mean? According to the paper, I understand that there is no chemical shift in NMR spectra of the samples, but there are differences in the chemical structure based on FTIR, XRD, TGA, and XPS, so I think “*no chemical transformation*” could be misunderstood or misinterpreted by the chemical community.
3. In **Introduction** (page 3/14), please provide references for the following sentences:
Paragraph 2: ... *the presence of either organic solvents such as ethanol, or surfactants.*^X
Paragraph 2: ... *instability of small surfactant based-particles are well-known drawbacks.*^X
Paragraph 3: ... *polymeric particles with more than 35% loading of α -mangostin are rare.*^X
4. In **Materials and Methods** (page 4/14), please add the following detail:
2-1; Paragraph 1: ... *Then water was slowly dropped into the solution and particles were formed.* If the addition of water into ethanolic solution was while stirring the ethanolic solution? If yes, what was the speed (i.e., rpm) of stirring?
2-1; Paragraph 2: Add NMR characterization detail here.
2-1; Paragraphs 3, 4, 5, and 6: Move these paragraphs to SI after Fig. S2.
5. In **Results and Discussion**:
Page 5/14: In explaining the effect of α -mangostin concentrations on particle sizes, the authors could simply mention that in low concentration, the hydrophilic surfaces on α -mangostin are more accessible, and consequently α -mangostin disperse well in aqueous environment, leading to smaller particles.
Page 6/14, Figure 1: I recommend: 1) moving **A4** and **B1** to SI for better data presentation, 2) changing Fig. 1 caption to “SEM images of MG particles prepared under various conditions: **A**) α -mangostin concentrations of 0.5% (A1), 1% (A2), and 5% (A3); **B**) ethanol

to water ratios of 0.5% (B2), 5% (B3), and 20% (B4); and C) 50% solvent to water ratio when the solvent is ethanol (C1), propanol (C2), and butanol (C3).”

Page 6/14: The authors reported dry particle sizes from SEM images; how they measured particle sizes? Any software was used? how many particles were counted? Add the detail to 2-1; Paragraph 2.

Page 6/14: In the explanation of bimodal distribution, I recommend the authors to consider the **boiling point** and **size** of organic solvents; which solvent has a lower boiling point and is smaller, so it is easier to escape and evaporate?

Page 7/14, Figure 2a: Fig. 2a is so small, and FTIR detail explained in the manuscript is hard to follow; please present Fig. 2a separately.

Page 7/14: In explaining FTIR data, the authors stated that “*The O-H stretching vibration of the original α - mangostin shows up as the double broad peaks at 3412.2 and 3246.3 cm^{-1}* ”; also they observed double peaks for C-O stretching of α - mangostin. Any explanation for these observation? These double peaks are associated with which type of stretching vibrations? Any previous literature with similar observation to refer to it?

Page 7/14: The authors stated that “*Since all the tested materials are pure α -mangostin with the same molecular structure, the differences in the FT-IR spectra imply different intermolecular interactions.*” **When the molecular structure is the same, how intermolecular interactions between same materials are different? Also, there are differences in functional group and structure based on FTIR, XRD, TGA, and XPS, so all means no change in molecular structure?**

Page 7/14, Figure 2c: Add materials names to this Figure.

Page 7/14, Figure 2 caption: Please correct “... of original α -mangostin crystalline solid, nMG-E (nanomangostin particles made in 50% aqueous ethanol, *blue* line), nMG-P (nanomangostin particles made in 50% aqueous propanol, *red* line) and ...” to “of original α -mangostin crystalline solid (*blue* line), nMG-E (nanomangostin particles made in 50% aqueous ethanol, *red* line), nMG-P (nanomangostin particles made in 50% aqueous propanol, *green* line) and ...”

Page 8/14, Figure 3: Based on HR-XPS of C 1s, higher binding energy features correspond to the oxidized species, and the particles have more oxidized moieties compared to original α -mangostin. Hence, I recommend the authors to move peak area table to SI.

6. In **SI**, list Tables and Figures in the order they appear in the manuscript.

Appendix B

In this work, the authors prepared water-dispersible α -mangostin particles using a straightforward approach of the slow addition of water into an ethanolic solution of α -mangostin without the use of surfactants or polymers. They also systematically studied the effect of α -mangostin concentration, ethanol/water ratio, as well as the type of organic solvent on the size of α -mangostin particles. They finally characterized particles using a variety of techniques (e.g., FTIR, XRD, TGA, XPS, SEM, DLS) and tested their anti-acne and anti-inflammatory activities.

This work is suitable for publication in RSOS, and it can be considered for publication after addressing the following comments and questions:

1. In **Title** (page 3/14), the authors used the word “*nanoparticles*”. Nanoparticles are routinely defined as particles with sizes in nanoscale (i.e., **between 1 and 100 nm**), exhibiting properties that are not found in their bulk counterparts. According to the paper, the average sizes of particles obtained from SEM images are **above 300 nm**, so I would suggest changing the word “*nanoparticles*” in the title and the whole manuscript to either “*colloidal particles*” or “*particles*”. I also suggest mentioning the average sizes of particles in the Abstract (page 3/14).

Response: Changes have been made accordingly in the revised manuscript.

2. In **Abstract** (line 8, page 3/14), the authors mention “*no chemical transformation*”; what does this mean? According to the paper, I understand that there is no chemical shift in NMR spectra of the samples, but there are differences in the chemical structure based on FTIR, XRD, TGA, and XPS, so I think “*no chemical transformation*” could be misunderstood or misinterpreted by the chemical community.

Response: We mean that there was no chemical reaction into new compound, only changes in molecular interaction among mangostin molecules. We have re-written the sentence.

3. In **Introduction** (page 3/14), please provide references for the following sentences: **Paragraph 2:** ... *the presence of either organic solvents such as ethanol, or surfactants.*X

Yes, added accordingly.

Paragraph 2: ... *instability of small surfactant based-particles are well-known drawbacks.*X

Yes, added accordingly.

Paragraph 3: ... *polymeric particles with more than 35% loading of α - mangostin are rare.*X

Yes, added accordingly

4. In **Materials and Methods** (page 4/14), please add the following detail: **2-1; Paragraph 1:** ...*Then water was slowly dropped into the solution and particles were formed.* If the addition of water into ethanolic solution was while stirring the ethanolic solution? If yes, what was the speed (i.e., rpm) of stirring?

The details have been added.

2-1; Paragraph 2: Add NMR characterization detail here.

Response: Moved accordingly

2-1; Paragraphs 3, 4, 5, and 6: Move these paragraphs to SI after Fig. S2.

Response: Moved accordingly

5. In Results and Discussion:

Page 5/14: In explaining the effect of α -mangostin concentrations on particle sizes, the authors could simply mention that in low concentration, the hydrophilic surfaces on α -mangostin are more accessible, and consequently α -mangostin disperse well in aqueous environment, leading to smaller particles.

Response: We have made the explanation more concise.

Page 6/14, Figure 1: I recommend: 1) moving **A4** and **B1** to SI for better data presentation, 2) changing Fig. 1 caption to "SEM images of MG particles prepared under various conditions: **A)** α -mangostin concentrations of 0.5% (A1), 1% (A2), and 5% (A3); **B)** ethanol to water ratios of 0.5% (B2), 5% (B3), and 20% (B4); and **C)** 50% solvent to water ratio when the solvent is ethanol (C1), propanol (C2), and butanol (C3)."

Response: Rearranged as suggested.

Page 6/14: The authors reported dry particle sizes from SEM images; how they measured particle sizes? Any software was used? how many particles were counted? Add the detail to 2-1; Paragraph 2.

Response: Details have been added.

Page 6/14: In the explanation of bimodal distribution, I recommend the authors to consider the **boiling point** and **size** of organic solvents; which solvent has a lower boiling point and is smaller, so it is easier to escape and evaporate?

Response: Added accordingly

Page 7/14, Figure 2a: Fig. 2a is so small, and FTIR detail explained in the manuscript is hard to follow; please present Fig. 2a separately.

Response: We have made Fig 2a bigger.

Page 7/14: In explaining FTIR data, the authors stated that "*The O-H stretching vibration of the original α - mangostin shows up as the double broad peaks at 3412.2 and 3246.3 cm^{-1}* "; also they observed double peaks for C-O stretching of α - mangostin. Any explanation for these observation? These double peaks are associated with which type of stretching vibrations? Any previous literature with similar observation to refer to it?

Response: Some explanation has been put in.

Page 7/14: The authors stated that "*Since all the tested materials are pure α -mangostin with the same molecular structure, the differences in the FT-IR spectra imply different intermolecular interactions.*"

When the molecular structure is the same, how intermolecular interactions between same materials are different? Also, there are differences in functional group and structure based on FTIR, XRD, TGA, and XPS, so all means no change in molecular structure?

Responses: Configurational differences. We have clarified this with the more specific wording.

Page 7/14, Figure 2c: Add materials names to this Figure.

Response: Added as suggested

Page 7/14, Figure 2 caption: Please correct "... of original α -mangostin crystalline solid, nMG-E (nanomangostin particles made in 50% aqueous ethanol, blue line), nMG-P (nanomangostin particles made in 50% aqueous propanol, red line) and ..." to "of original α -mangostin crystalline solid (blue line), nMG-E (nanomangostin particles made in 50% aqueous ethanol, red line), nMG-P (nanomangostin particles made in 50% aqueous propanol, green line) and ..."

Response: Already made it clearer as suggested.

Page 8/14, Figure 3: Based on HR-XPS of C 1s, higher binding energy features correspond to the oxidized species, and the particles have more oxidized moieties compared to original α -mangostin. Hence, I recommend the authors to move peak area table to SI.

In **SI**, list Tables and Figures in the order they appear in the manuscript.

Response: Changed accordingly.

Appendix C

Title: Water-dispersible unadulterated α -mangostin particles for biomedical applications

Manuscript ID: RSOS-200543.R1

Reviewer comments to Author:

Reviewer: 1

Comments to the Author(s)

Looks great! Only 2 minor comments:

1. Fig. 2 caption (page 4): change it to "Structural analyses of original α -mangostin crystalline solid, MG-E (mangostin particles made in 50% aqueous ethanol), MG-P (mangostin particles made in 50% aqueous propanol) and MG-B (mangostin particles made in 50% aqueous butanol): (a) IR spectra, (b) X-ray diffraction patterns and (c) thermograms."

Response: Changed accordingly.

2. Fig. 4 caption (page 6): This must be Fig. 4 (not Fig. 2), correct accordingly.

Response: Caption of Fig 4 is definitely for Fig 4.